# Non-target effects of a glyphosate-based herbicide on Common toad larvae (*Bufo bufo*, Amphibia) and associated algae are altered by temperature

Fabian Baier[1], Edith Gruber[1], Thomas Hein[2,3],
Elisabeth Bondar-Kunze[2,3], Marina Ivanković[2], Axel Mentler[4],
Carsten A. Brühl[5], Bernhard Spangl[6] and Johann G. Zaller[1]

[1] Institute of Zoology, University of Natural Resources and Life Sciences Vienna (BOKU), Vienna, Austria
[2] Institute of Hydrobiology and Aquatic Ecosystem Management, University of Natural Resources and Life Sciences Vienna (BOKU), Vienna, Austria
[3] WasserCluster Lunz–Biologische Station GmbH, Lunz am See, Austria
[4] Institute of Soil Research, University of Natural Resources and Life Sciences Vienna (BOKU), Vienna, Austria
[5] Institute for Environmental Sciences, Universität Koblenz-Landau, Landau, Germany
[6] Institute of Applied Statistics and Computing, University of Natural Resources and Life Sciences Vienna (BOKU), Vienna, Austria

Corresponding author
Johann G. Zaller,
johann.zaller@boku.ac.at

## ABSTRACT

**Background:** Glyphosate-based herbicides are the most widely used pesticides in agriculture, horticulture, municipalities and private gardens that can potentially contaminate nearby water bodies inhabited by amphibians and algae. Moreover, the development and diversity of these aquatic organisms could also be affected by human-induced climate change that might lead to more periods with extreme temperatures. However, to what extent non-target effects of these herbicides on amphibians or algae are altered by varying temperature is not well known.

**Methods:** We studied effects of five concentrations of the glyphosate-based herbicide formulation Roundup PowerFlex (0, 1.5, 3, 4 mg acid equivalent glyphosate $L^{-1}$ as a one time addition and a pulse treatment of totally 4 mg a.e. glyphosate $L^{-1}$) on larval development of Common toads (*Bufo bufo*, L.; Amphibia: Anura) and associated algae communities under two temperature regimes (15 vs. 20 °C).

**Results:** Herbicide contamination reduced tail growth (−8%), induced the occurrence of tail deformations (i.e. lacerated or crooked tails) and reduced algae diversity (−6%). Higher water temperature increased tadpole growth (tail and body length (tl/bl) +66%, length-to-width ratio +4%) and decreased algae diversity (−21%). No clear relation between herbicide concentrations and tadpole growth or algae density or diversity was observed. Interactive effects of herbicides and temperature affected growth parameters, tail deformation and tadpole mortality indicating that the herbicide effects are temperature-dependent. Remarkably, herbicide-temperature interactions resulted in deformed tails in 34% of all herbicide treated tadpoles at 15 °C whereas no tail deformations were observed for the herbicide-free control at 15 °C or any tadpole at 20 °C; herbicide-induced mortality was higher at 15 °C but lower at 20 °C.

**PeerJ** _________________________________

**Discussion:** These herbicide- and temperature-induced changes may have decided effects on ecological interactions in freshwater ecosystems. Although no clear dose-response effect was seen, the presence of glyphosate was decisive for an effect, suggesting that the lowest observed effect concentration (LOEC) in our study was 1.5 mg a.e. glyphosate $L^{-1}$ water. Overall, our findings also question the relevance of pesticide risk assessments conducted at standard temperatures.

# INTRODUCTION

With almost a third of their species threatened, amphibians are the most vulnerable vertebrate group on earth (_International Union for Conservation of Nature, 2004_). Amphibian declines and extinctions are caused by multiple factors and their interaction: alien species, over-exploitation and land use change, global change, increased use of pesticides and other toxic chemicals and infectious diseases (_Aldrich et al., 2016_; _Blaustein et al., 2011_; _Brühl et al., 2013_; _Collins & Storfer, 2003_). In their natural habitats, especially anuran amphibians are always associated with algae communities that constitute an important source of nutrition during larval development; however, algae are likewise affected by similar environmental factors as amphibians (_Whitehead et al., 2009_; _Wilson et al., 2003_). The current study examined the effects of a glyphosate-based herbicide on growth and development of larvae of Common toads (_Bufo bufo_ L.) and on the composition of the associated algal communities. Additionally, as climate change scenarios predict more frequent extreme temperature events (_IPCC, 2013_), we also assessed whether different water temperatures alter potential herbicide effects. Moreover, toad spawning ponds can vary considerably in their depths and in their water temperature. Globally, Common toads are mainly distributed in Europe living in habitats from lowland to mountain areas up to 2,500 m a.s.l. (_Arnold & Ovenden, 2002_; _Recuero et al., 2012_). Since Common toads frequently spawn in ponds located near agriculturally used areas or because they need to cross these landscapes during their seasonal migrations from the spawning ponds to woody overwintering sites, they are frequently exposed to pesticides (_Berger, Graef & Pfeffer, 2013_; _Lenhardt, Brühl & Berger, 2015_). Common toads were selected as a study organism also for ethical reasons because they are among the least threatened of the generally endangered animal class of Amphibia.

Glyphosate-based herbicides are the most often and broadly used pesticides in agriculture, horticulture, forestry, vineyards, municipalities and private gardens. Generally, glyphosates' mode of action inhibits an enzyme only known to plants and some microorganisms; therefore, and due to the half-life of 7–14 days in water it is considered only slightly toxic to amphibians (_Franz, Mao & Sikorski, 1997_; _Giesy, Dobson & Solomon, 2000_). Most glyphosate-based herbicides are not meant to be used in

or close to aquatic environments, however, contamination of water bodies through drift, surface runoff and direct overspray has been reported (*Battaglin et al., 2009*; *Berger, Graef & Pfeffer, 2013*; *Peruzzo, Porta & Ronco, 2008*). Nevertheless, several studies report detrimental effects of glyphosate-based herbicide formulations to amphibians during their aquatic and terrestrial life stages (*Baylis, 2000*; *Brühl, Pieper & Weber, 2011*; *Duke & Powles, 2008*; *Fryday & Thompson, 2012*; *Relyea, 2011*; *Tsui & Chu, 2003*). Glyphosate-based herbicides tested in concentrations ranging between 0.021 mg acid equivalent (a.e.) $L^{-1}$ to 6.0 mg a.e. $L^{-1}$ affected growth and development of North American larval anurans (*Edge et al., 2014*; *Lanctôt et al., 2014*; *Navarro-Martín et al., 2014*; *Relyea, 2004*), showed other sub-lethal morphological and behavioral effects (*Cauble & Wagner, 2005*; *Mann & Bidwell, 1999*; *Moore, Chivers & Ferrari, 2015*; *Muñoz, Rojas & Bautista, 2014*) and even lead to acute mortality when exposed to high doses (*Fuentes et al., 2011*; *Moore et al., 2012*; *Relyea, 2005b*; *Williams & Semlitsch, 2010*). However, concerning effects of glyphosate-based herbicides on native European amphibian species only very little is known (*Wagner & Lötters, 2013*). Moreover, it has been shown that herbicide formulations, i.e. the products that are actually used in the field, are more detrimental to tadpoles and other aquatic organisms than the active ingredient glyphosate itself (*Folmar, Sanders & Julin, 1979*; *Howe et al., 2004*; *Relyea, 2005b*). However, unfortunately these adjuvants are usually not declared and considered company secrets (*Mullin et al., 2016*; *Wagner et al., 2013*). Besides amphibians, several studies also show that freshwater microalgae are particularly vulnerable to glyphosate-based herbicides because of their physiological and biochemical similarity with terrestrial plants (*Annett, Habibi & Hontela, 2014*; *Tsui & Chu, 2003*) mainly by initiating oxidative stress in algae (*Annett, Habibi & Hontela, 2014*).

Temperature has long been known to play an important role in the physiology and ecology of tadpoles (*Katzenberger et al., 2014*; *Ultsch, Bradford & Freda, 1999*) and algae (*Butterwick, Heaney & Talling, 2005*; *Schabhüttl et al., 2013*). Although climate warming is not considered to be directly lethal to amphibians, increasing water temperatures in amphibian breeding ponds may accelerate larval development and hence reduce the duration of herbicide exposure for these species (*Li, Cohen & Rohr, 2013*; *Rohr & Palmer, 2013*). Potential interactions between herbicides and temperature are of particular interest as human-induced climate change can increase mean temperatures and will most likely also lead to weather events with more extreme temperatures (*IPCC, 2013*). On the other hand, pesticide use is also expected to rise with rising temperatures because certain pests and diseases are expected to benefit from climate change (*Kattwinkel et al., 2011*). Interactions of pesticide and global warming effects have been predicted and outlined as a major point of concern for wildlife (*Köhler & Triebskorn, 2013*; *Rohr & Palmer, 2013*). Only very few studies investigated combined effects of herbicides and temperature on amphibians (*Baier et al., 2016*; *Rohr, Sesterhenn & Stieha, 2011*). These studies show that increasing temperatures generally do not enhance the toxicity of the herbicide, but rather temperature ameliorates the adverse effects of the herbicide by accelerating development and reducing the duration of herbicide exposure (*Baier et al., 2016*; *Rohr, Sesterhenn & Stieha, 2011*).

Thus, the objectives of the current study were to assess whether (i) herbicide concentrations affect the development of Common toad larvae and associated algae communities and (ii) temperature alters potential herbicide effects. To investigate these objectives we conducted a climate chamber experiment using a full factorial design with five herbicide concentrations at two temperature levels over 24 days. We hypothesized that a higher temperature will accelerate the growth and development of tadpoles, hence reducing the duration of their exposure to the herbicide. When higher temperature reduces exposure to the herbicide more than it enhances detrimental effects of the herbicide, then its net effect could be beneficial. Regarding herbicide effects we also hypothesized that increasing temperatures would increase detrimental effects of the herbicide with increasing concentrations because multiple stressors might decrease resources of amphibians for detoxification and/or temperature regulation (*Noyes et al., 2009*; *Rohr, Madison & Sullivan, 2003*).

## MATERIAL AND METHODS

### Experimental setup

The study was conducted between April 14 and May 9, 2015 in a laboratory at the University of Natural Resources and Life Sciences Vienna (BOKU), Austria. This study was carried out in strict accordance with the recommendations in the Austrian animal experimentation law (Tierversuchsgesetz 2012, BGBl. I Nr. 114/2012) with a permission from the Austrian Federal Ministry of Science, Research and Economy (permit number BMWFW-66.016/0002-WF/V/3b/2015). Anesthetic MS222 (Triacin, Sandoz, Switzerland) was used when necessary to ameliorate animal suffering. The experiment was carried out using a full-factorial design, with herbicide concentration (five levels) and temperature (two levels) as factors. Two climate chambers set at different temperatures (15 and 20 °C) with a 12 h light/dark-cycle were used to set up the two temperature conditions. Light intensity of both climate chambers was measured using a Lux Meter (Voltcraft LX-1108 Lux-Meter; Conrad electronics, Hirschau, Germany) resulting in $5,575 \pm 106$ lux (mean $\pm$ SD) in both chambers. We established the following herbicide concentrations: 0, 1.5, 3, 4 mg a.e. $L^{-1}$ (acid equivalent; which equals 0, 3.13, 6.25, 8.33 $\mu$l $L^{-1}$ of the Roundup PowerFlex® formulation) from the start and 4 mg a.e. $L^{-1}$ applied as a pulse treatment starting initially with 1.5 mg a.e. $L^{-1}$ and adding another 1.5 mg a.e. $L^{-1}$ on day 5 and 1 mg a.e. $L^{-1}$ (2.08 $\mu$l $L^{-1}$) on day 11 after starting the experiment, respectively. In the field, glyphosate concentrations in amphibian breeding ponds are generally highly variable depending on potential pesticide drift or input through leaching by water or soil erosion. Worst-case expected environmental concentration of 7.6 mg a.e. glyphosate $L^{-1}$ have been calculated (*Wagner et al., 2013*), however during most of the year concentrations in spawning ponds are likely to be lower (*Peruzzo, Porta & Ronco, 2008*; *Struger et al., 2008*; *Trumbo, 2005*). Five 25-L plastic canisters were used to prepare herbicide solutions. First each canister was filled with 22 L of coal filtered tap water, then the glyphosate-based herbicide Roundup PowerFlex (containing: 588 g $L^{-1}$ potassium salt of glyphosate, which equals 480 g $L^{-1}$ glyphosate acid; Monsanto Europe S.A./N.V., Belgium) was added in the desired concentrations using

a micro-pipette. This Roundup formulation is available in Europe since 2014 and admitted for usage in agriculture, horticulture, forestry, vineyards, municipalities and private gardens until 2022 (*AGES, 2015*). Afterwards the solutions were mixed thoroughly by shaking and turning the canisters upside down. Four liters of pure coal-filtered water (in case of the control) or the mixtures were then filled into each experimental unit, i.e. polypropylene plastic tubs (volume: 5-L, length: 28 cm, width: 19 cm, height: 14 cm), accounting for five replicates per treatment. Each climate chamber contained 25 tubs, making up a total of 50 tubs for the two temperature treatments.

Common toad (*Bufo bufo* L.) spawn containing approx. 750 eggs from different clutches was hand-collected on 14 April 2015 from a pond in Neuwaldegg, Vienna (48°14′26.31″N; 16°16′34.945″E; altitude 290 m a.s.l.). Toad eggs were sampled with permission of the pond owner (Vienna Municipal Department 49—Forestry, permit number MA49-808754/2014/3) and the respective authority for nature conservation (Vienna Municipal Department 22—Environmental Protection, permit number MA22-1629490/2014). Five randomly chosen eggs of the whole spawn were added into each experimental tub, accounting for a total of 250 eggs used for this experiment.

## Measurements and analyses

Larval development was determined according to *Gosner (1960)* using a binocular microscope (Nikon SMZ 745; Nikon, Tokyo, Japan). Conforming to this scheme the development of anurans is assigned to 46 stages (Gosner stage (Gs)), from undivided eggs (Gs 1) to completely metamorphosed individuals (Gs 46). At the start of our experiment all tadpoles were in Gs 8. As additional morphological parameters we measured bl, body width (bw) and tail length (tl) from photographs taken of tadpoles from all treatments at May 8, when the maturity of larval amphibians reached Gs 30 and Gs 35 at 15 and 20 °C, respectively. Measurements on photographs were conducted using the free image analysis software ImageJ (version 1.48v, Wayne Rasband, National Institute of Health, USA: http://imagej.nih.gov/ij). Body condition (bc) was calculated as body length-to-width ratio. Tail deformations (*Bantle et al., 1998*; *Cooke, 1981*) were assessed when taking the tadpoles out into Petri dishes for morphological measurements at the end of the experiment. A clear distinction could be made between undeformed (Fig. 1A), lacerated (Fig. 1B) or curved tails (Fig. 1C); only curved tails were considered deformed (*Cooke, 1981*) as lacerated tails could also originate from injuries by intraspecific competition.

Each tub was inspected daily for dead individuals. All larvae were fed ad libitum with same amounts of ground fish food (Tetramin) after reaching Gs 22. When most of the 20 °C-treated tadpoles reached Gs 35 (indentations 1–2 of the hind limbs develop at this stage) the experiment was terminated. Water samples of all tubs (also the 15 °C ones) were taken after thoroughly mixing with a glass stirring rod using a 100 ml plastic beaker; samples were preserved by adding five drops of Lugol iodine solution.

In these water samples algae species composition was examined. Algae were introduced into the experimental units via the tadpole eggs; as eggs were randomly assigned to the treatments we considered this as a similar initial situation for all treatments. For algae

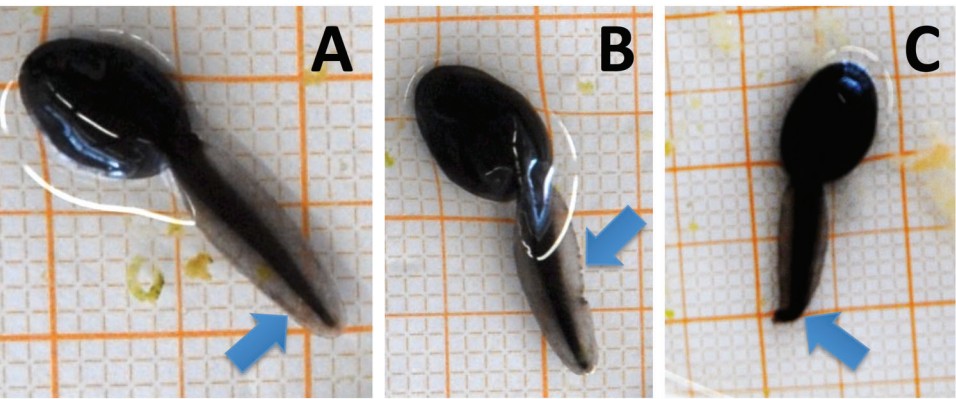

**Figure 1 (A) Tadpoles observed in the current experiment with undeformed, (B) lacerated and (C) curved tail tips.** Only curved tail tips were considered deformed and included in the analyses.

species determination, the phytoplankton samples were gently mixed by inverting the sample bottle. Initially a preliminary scan was made of a settled 5 ml sample in order to determine the volume to be used for the analyses. For the qualitative analysis at least 5 ml of the sample were analyzed. As a high number of algal cells were present, 1 ml was loaded into a sedimentation chamber of appropriate volume for quantitative analysis. The chamber was topped with a round glass top plate and algae were allowed to settle onto the base of the settling chamber. The method consists of two parts—analysis of non-diatom phytoplankton ("soft algae") and analysis of diatoms. For the non-diatom phytoplankton analysis, algal cells were enumerated in a settling chamber using an inverted microscope at 400× magnification. Only living forms (i.e. chloroplast containing cells) were counted. At this magnification diatoms are enumerated as a group. For diatom analyses, the samples were pretreated with strong oxidants, cleaned and embedded on glass slides in Naphrax. Actual identification and enumeration of diatoms was done under oil immersion (1,000×) and for the quantitative analyses a minimum of 500 frustules was counted (2 frustules = 1 diatom cell) per sample (slide). For the quantitative analyses a minimum of 30 microscopic fields is required, including no less than 300 cells. If 300 cells were not observed within the 30 microscopic fields, enumeration was continued until at least 300 were counted. The area counted was recorded to calculate cell number per volume. Algal taxa were identified to the lowest taxonomic rank possible. The taxonomical identification was done using standard literature for species determination (*Hindák et al., 1978*; *Krammer & Lange-Bertalot, 1986*; *Krammer & Lange-Bertalot, 1988*; *Krammer & Lange-Bertalot, 1991a*; *Krammer & Lange-Bertalot, 1991b*). Species names were checked at the algae base database (*AlgaeBase, 2015*). Quantitative assessment was conducted according to *Utermöhl (1958)*. The density is presented as cell number $L^{-1}$.

Water temperature and water oxygen concentration were measured using an Oxymeter (WTW Oxi 90; WTW GmbH, Weilheim, Bavaria, Germany) and pH was measured using a pH-Meter (WTW pH-95; WTW GmbH, Weilheim, Bavaria, Germany). Solved oxygen content, pH-value and temperature of the water in the tubs was first measured on April 14, shortly before introducing the eggs. Henceforth, measurements were performed

every third day until May 8. A data logger measuring temperature every 15 min was placed in the middle section of each climate chamber at the start of the experiment. The mean temperature during the experiment was 14.4 vs. 18.3 °C for the 15 vs. 20 °C treatment, respectively.

Water samples (500 ml) of each tub were taken at the start and end of the experiment and immediately stored in a freezer at −18 °C before they were analysed for glyphosate and its main metabolite aminomethylphosphonic acid (AMPA) in the laboratories of the BOKU Department of Forest and Soil Sciences using a HPLC-MS/MS method (*Popp et al., 2008*; *Todorovic et al., 2013*).

## Statistical analyses

All statistical analyses were carried out using the open-access software R (version 3.2.1, The R Foundation for Statistical Computing: http://www.r-project.org); α was set at 0.05. Shapiro-Wilks tests and QQ-plots were used to check normality of data. Percentage mortality was arcsin-transformed prior to analysis. To examine whether herbicide concentrations and temperature affected toad larvae morphology, analyses of covariance (ANCOVA) with pH, measured water temperature and oxygen levels as co-variables were conducted. Effects of herbicide and temperature on pH and dissolved oxygen were tested using a two-way ANOVA. To analyze effects of herbicide, temperature, pH and dissolved oxygen on algae density and diversity a different ANCOVA model was used as samples were only taken at one point of time. To assess the effects of algae density and diversity on bl, bw, tl and bl to bw ratio of tadpoles a one-way ANCOVA was conducted. When significant effects were given by the ANCOVA ($P < 0.05$) a Tukey HSD test was conducted for mean comparisons of herbicide and temperature levels. Water pH and oxygen concentrations were used as covariates in all ANCOVA models. Correlations were tested using Pearson's test. Values in the text are means ± standard deviation.

## RESULTS

Herbicide contamination significantly affected tadpole tl at 15 °C but not at 20 °C, occurrence of tail deformations only at 15 °C and algae diversity (Shannon- and evenness-index) mainly at 20 °C (Figs. 2 and 3; Table 1). Herbicide concentrations neither affected larval development, bl, bw, bc, mortality of tadpoles or algae density (Figs. 2 and 3; Table 1).

Temperature significantly affected larval growth parameters (bl, bw, tl and bc), larval development (Gs) and occurrence of tail deformations of tadpoles as well as algae diversity (Shannon- and evenness-index); however it did not affect tadpole mortality or algae density (Figs. 2 and 3; Table 1). Algal communities contained between 25–43 species, with the highest species number in the 15 °C treatment (Fig. 3). Diatom algae dominated in species number and abundance in all mesocosms.

Interactions between herbicide and temperature significantly affected larval tl, bl, bw, tadpole mortality and occurrence of tail deformations, as well as evenness of algal communities (Figs. 2 and 3; Table 1). This significant herbicide × temperature interaction led to the lowest tl (9.36 ± 0.64 mm) in the 4 mg a.e. L$^{-1}$ pulse treatment at 15 °C and the

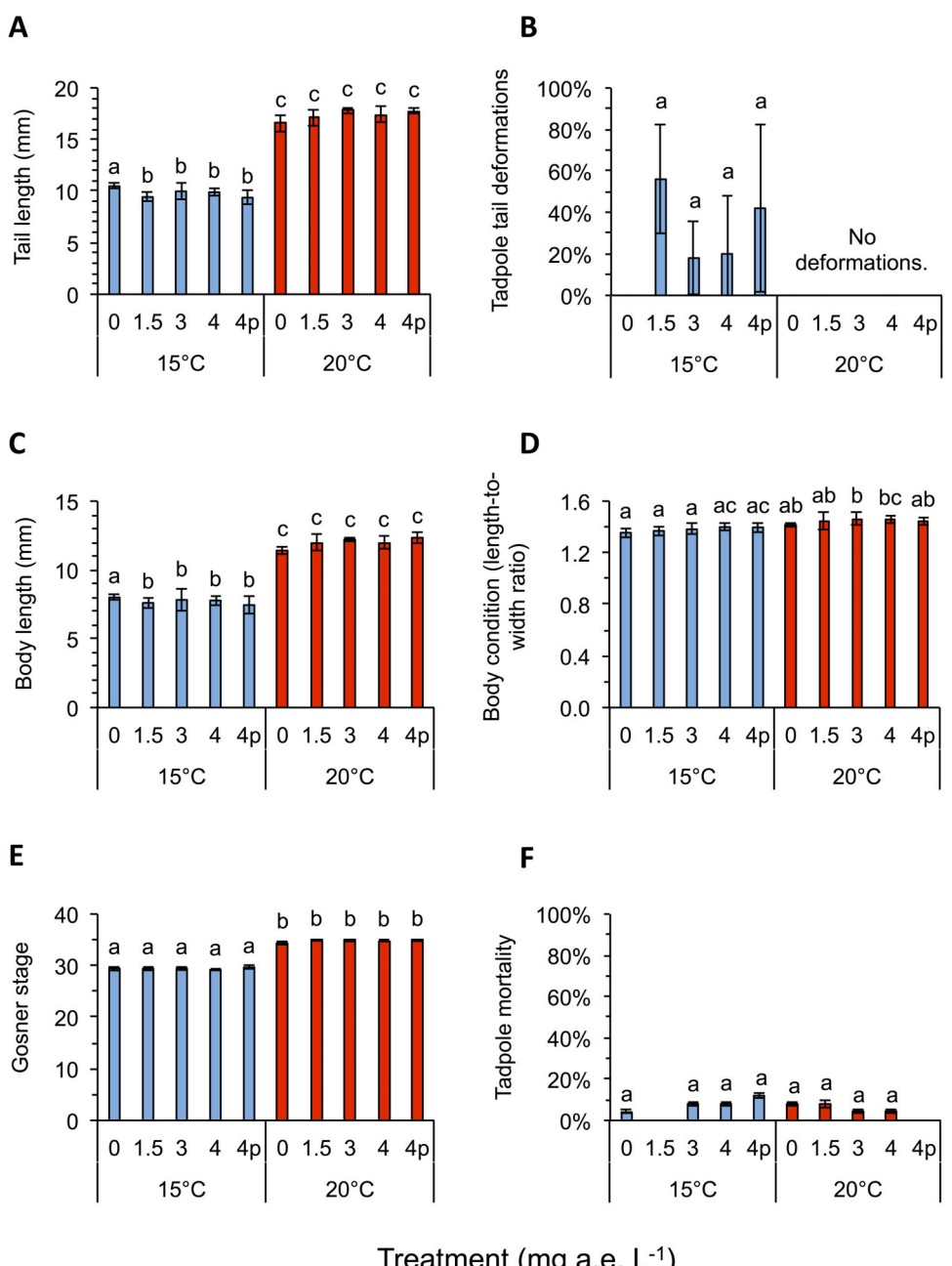

**Figure 2** **(A) Tail length, (B) tail deformation, (C) body length, (D) body condition, (E) development and (F) mortality of Common toad tadpoles in response to different herbicide treatments (mg a.e. L$^{-1}$) at 15 or 20 °C, measured 24 days after experiment start.** Mean ± SD, n = 5. Means with different letters are significantly different (Tukey HSD, P < 0.05).

longest tails (17.79 ± 0.31 mm) at 3 mg a.e. L$^{-1}$ under 20 °C (Fig. 2A). At 15 °C tadpoles in tubs contaminated with the herbicide had averaged across herbicide concentrations 7.8% shorter tails compared to the control treatment (Fig. 2A).

Twenty four days after experiment start, tadpoles of the 15 °C chamber were on average in Gs 29, whereas tadpoles at 20 °C were in Gs 35 (Fig. 2E). The bc was highly significantly

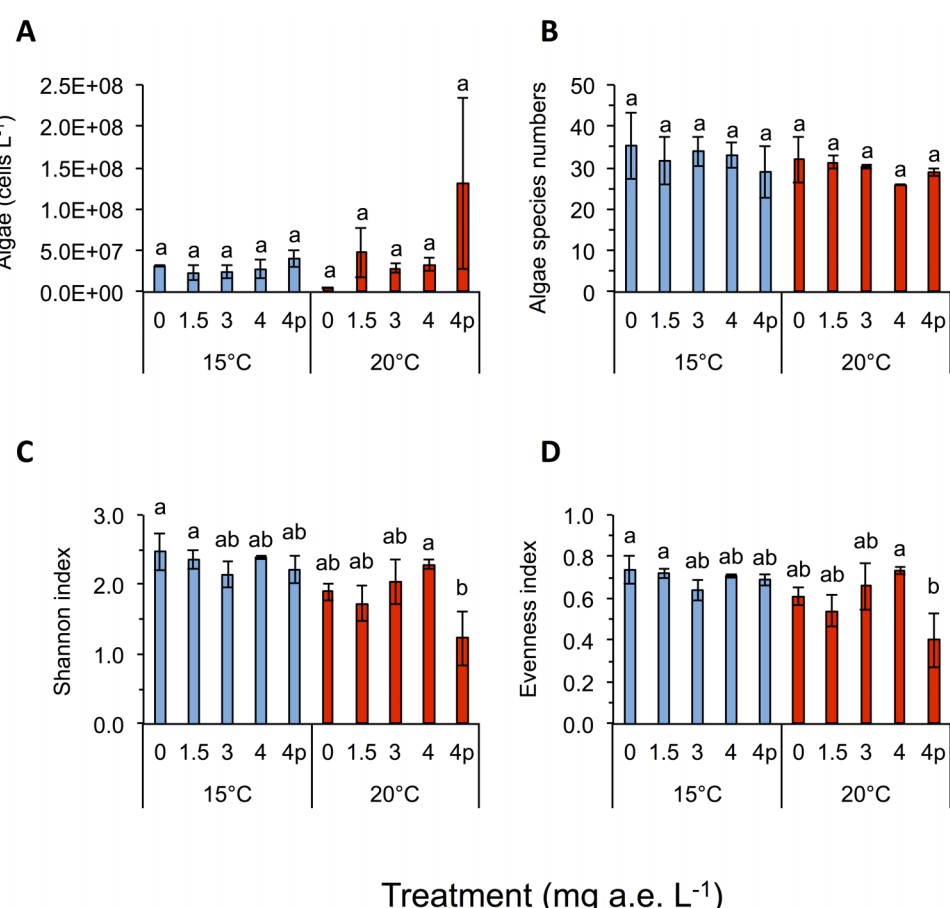

**Figure 3** **(A) Algae abundance, (B) number of species (C) diversity depicted using Shannon- and (D) evenness-index in response to different herbicide treatments (mg a.e. L$^{-1}$) at 15 or 20 °C.** Mean ± SD, 24 days after experiment start, n = 3. Means with different letters are significantly different (Tukey HSD, P < 0.05).

affected by temperature, being reduced by 4.2% at 15 °C compared to 20 °C, on average across herbicide concentrations (Fig. 2D; Table 1). The highest bc (1.46 ± 0.05) was found at 3 mg a.e. L$^{-1}$ under 20 °C, the lowest bl-to-bw ratio (1.36 ± 0.04) at 0 mg a.e. L$^{-1}$ under 15 °C. Tadpole bl and bw also differed significantly under different temperature regimes. Body length at 15 °C was lower by 35.4% on average compared to 20 °C; whereas bw at 15 °C was lower by 32.4% on average compared to 20 °C (Fig. 2C). At 15 °C bl at 0 mg a.e. L$^{-1}$ (8.0 ± 0.18 mm) was significantly higher compared to the other herbicide treatments (7.67 ± 0.55 mm); at 20 °C average bl (11.99 ± 0.39 mm) was unaffected by herbicide concentrations (Fig. 2C). At 15 °C bw at 4p mg a.e. L$^{-1}$ (5.33 ± 0.33 mm) was significantly lower compared to the other herbicide concentrations (5.67 ± 0.29 mm); at 20 °C bw (8.29 ± 0.39 mm) was unaffected by herbicide concentrations.

At the end of the experiment on average 34.0 ± 14.8% of the tadpoles at 15 °C showed deformed tails, while none at 20 °C had deformed tails (Fig. 2B). Most deformed tadpoles (56.0 ± 26.0%) were found at 1.5 mg a.e. L$^{-1}$, however, no relation to herbicide concentration was found. No deformed tadpoles were found in the control group without

**Table 1** ANCOVA results for the effects of herbicide concentrations (0, 1.5, 3, 4, 4p mg a.e. $L^{-1}$) and temperature (15 vs. 20 °C) on growth stages, body length (bl), body width (bw), tail length, body condition (bl/bw), mortality and occurrence of tail deformations of Common toad larvae and diversity of associated algal communities.

| Variable | Herbicide concentrations | | Temperature | | Herbicide × Temperature | |
|---|---|---|---|---|---|---|
| | F | P | F | P | F | P |
| **Tadpoles** | | | | | | |
| Growth stages (Gosner) | 0.642 | 0.633 | 150.521 | **< 0.001** | 0.079 | 0.989 |
| Body length (mm) | 1.200 | 0.317 | 837.846 | **< 0.001** | 4.369 | **0.003** |
| Body width (mm) | 0.882 | 0.478 | 584.778 | **< 0.001** | 2.893 | **0.027** |
| Tail length (mm) | 4.113 | **0.004** | 891.085 | **< 0.001** | 4.851 | **0.001** |
| Body condition (bl/bw) | 1.413 | 0.236 | 17.040 | **< 0.001** | 2.303 | 0.065 |
| Mortality (%) | 0.401 | 0.807 | 0.760 | 0.386 | 2.821 | **0.030** |
| Tail deformation (%) | 2.792 | **0.040** | 5.236 | **0.028** | 3.077 | **0.027** |
| **Algae** | | | | | | |
| Algal density (cells $L^{-1}$) | 2.408 | 0.090 | 4.144 | 0.058 | 1.632 | 0.212 |
| Algal diversity (Shannon) | 5.022 | **0.007** | 9.888 | **0.006** | 2.761 | 0.062 |
| Algal diversity (Evenness) | 4.810 | **0.009** | 9.562 | **0.007** | 3.756 | **0.023** |

**Note:**
Significant results in bold.

herbicide addition. Whenever deformations occurred the tip of their tails was crooked in one direction (Fig. 1C).

Mortality was unaffected by herbicide treatment or temperature, but significantly affected by their interaction: on average, 6.4 ± 0.8% of larvae died at 15 °C vs. 5.6 ± 0.9% at 20 °C (Fig. 2F; Table 1). The highest mortality rate under 15 °C was found at 4p mg a.e. $L^{-1}$ (12.0 ± 1.1% mortality), whereas under 20 °C it was found at 0 mg a.e. $L^{-1}$ (8.0 ± 1.1% mortality); mortality of control at 15 °C was 4.0 ± 0.9%. No larvae died at 1.5 mg a.e. $L^{-1}$ under 15 °C and at 4p mg a.e. $L^{-1}$ under 20 °C (Fig. 2F).

Algal species of four different families were observed in the experiment, namely: Chlorophyceae (six spp.), Trebouxiophyceae (two spp.), Xanthophyceae (one spp.) and Bacillariophyceae (34 spp.). Algae Shannon diversity index under 15 °C was across treatments 20.7% higher than at 20 °C (Fig. 3C). At 15 °C the highest Shannon index was found at 0 mg a.e. $L^{-1}$ (2.48 ± 0.27), whereas under 20 °C the highest Shannon index was observed at 4 mg a.e. $L^{-1}$ (2.29 ± 0.06; Fig. 2). The lowest Shannon index was seen at 3 mg a.e. $L^{-1}$ (2.14 ± 0.18) under 15 °C and at 4p mg a.e. $L^{-1}$ (1.23 ± 0.39) under 20 °C (Fig. 3C). Algal density was unaffected by herbicide concentrations or temperature (Table 1). Averaged across concentrations we observed 33 species and 2.9E + 07 ± 7.8E + 06 algae cells $L^{-1}$ at 15 °C vs. 30 species and 4.9E + 07 ± 3.0E + 07 algae cells $L^{-1}$ at 20 °C (Figs. 3A and 3B).

**Table 2 ANCOVA results for the effects of water pH and oxygen concentration and time on development, body length, body width, tail length, body condition, mortality and tail deformation of *Bufo bufo* tadpoles and diversity of associated algae communities.**

| Variable | Water pH | | Water oxygen | |
|---|---|---|---|---|
| | F | P | F | P |
| **Tadpoles** | | | | |
| Growth Stage (Gosner) | 2.264 | 0.133 | 0.005 | 0.942 |
| Body length (mm) | 0.640 | 0.426 | 3.689 | 0.058 |
| Body width (mm) | 0.365 | 0.548 | 5.063 | **0.027** |
| Tail length (mm) | 0.017 | 0.898 | 14.450 | **< 0.001** |
| Body condition (bl/bw) | 2.388 | 0.126 | 0.044 | 0.835 |
| Mortality (%) | 0.118 | 0.732 | 0.288 | 0.592 |
| Tail deformation (%) | 0.109 | 0.744 | 0.302 | 0.586 |
| **Algae** | | | | |
| Algal density (cells L$^{-1}$) | 1.352 | 0.261 | 1.635 | 0.218 |
| Algal diversity (Shannon) | 0.361 | 0.556 | 0.375 | 0.548 |
| Algal diversity (Evenness) | 0.837 | 0.373 | 1.185 | 0.291 |

Note:
Significant results in bold.

Herbicide concentrations affected water pH and water oxygen concentrations, temperature treatment reduced water oxygen concentration (data provided in Supplemental Information S1). At 15 °C, water pH in the control (8.34 ± 0.13) was significantly lower than at treatments with herbicide addition (8.64 ± 0.48), at 20 °C control showed the lowest value (8.27 ± 0.07), whereas the pulse treatment had the highest (8.55 ± 0.41). Across temperatures water oxygen concentration increased with increasing herbicide concentrations. The lower temperature treatment showed higher oxygen values throughout all herbicide concentrations, the highest being 13.58 ± 4.50 mg L$^{-1}$ for the 4 mg a.e. L$^{-1}$ treatment at 15 °C, whereas the control at 20 °C had the lowest average oxygen value with 9.24 ± 0.68 mg L$^{-1}$. No significant effect of herbicide by temperature interaction on water pH or oxygen was observed. The temperature of the climate chamber had a significant influence on water temperature levels of the tubs, which differed by 3.9 °C. The mean temperature of the 15 and 20 °C treatment was 14.4 ± 1.3 °C and 18.3 ± 1.3 °C, respectively. Tadpole bw and tl were significantly affected by water oxygen concentration; however, they not affected by water pH (Table 2). Although bw was significantly affected by water oxygen concentration, those parameters were not significantly correlated. Tail deformation was not affected by pH or dissolved oxygen content (Table 2).

Neither algal density nor algal diversity showed any significant effects on bl, bw, tl and bc, although algal density had a marginal effect on tadpole bl (F = 4.096, P = 0.060).

Results of glyphosate and AMPA analyses confirmed the indented concentrations at the start of the experiment: < 5.0, 1,557.5 ± 33.6, 3,002.0 ± 6.3, and 4,050.7 ± 300.7 μg glyphosate L$^{-1}$ and generally < 2.5 μg AMPA L$^{-1}$ for the 0, 1.5, 3, 4 mg L$^{-1}$ one time herbicide application, respectively. Glyphosate and AMPA for the first 1.5 mg L$^{-1}$ addition

of the 4p mg L$^{-1}$ treatment was with 1,764.3 ± 61.8 µg glyphosate L$^{-1}$ higher than intended. At the end of the experiment measured concentrations across temperature treatments were: < 5.0, 1,491.4 ± 148.6, 2,923.6 ± 124.8, 3,979.6 ± 193.7, 6,568.3 ± 1,490.3 µg glyphosate L$^{-1}$ and < 2.5, 34.6 ± 13.1, 66.0 ± 35.6, 130.7 ± 16.2, 201.4 ± 29.8 µg AMPA L$^{-1}$, for the 0, 1.5, 3, 4 mg L$^{-1}$ one time herbicide and 4 mg L$^{-1}$ pulse treatment, respectively.

## DISCUSSION

Our study showed for the first time for Common toads that potential contaminations of spawning ponds with the glyphosate-based herbicide Roundup PowerFlex can reduce the tail growth of toad larvae, lead to tail deformations and reduce the diversity of associated algal communities. Increased water temperature accelerated tadpole development and growth and reduced algal diversity. Moreover, significant interactions between herbicide contamination and temperature affected most morphological parameters (bl, bw, tl, bc, mortality, tail deformation) and algal diversity (evenness-index), suggesting that the herbicide effects are temperature-specific. However, we found no clear indications of a relation between herbicide concentration and tadpole growth parameters or algae diversity or density. However, the presence of glyphosate was decisive for an effect, suggesting that the lowest observed effect concentration (LOEC) was 1.5 mg a.e. glyphosate L$^{-1}$ water.

Overall, our findings for a European amphibian species corroborate former studies showing effects of different Roundup formulations on the growth of mainly North-American amphibian species (*Edge et al., 2014*; *Howe et al., 2004*; *Lanctôt et al., 2014*; *Navarro-Martín et al., 2014*; *Relyea, 2004*; *Relyea, 2012*; *Wojtaszek et al., 2004*). Our results of 4.2% decrease in tl of *B. bufo* tadpoles in tubs containing Roundup PowerFlex (15 °C) compared to the control is in line with findings of a previous study where we tested effects of Roundup LB Plus on *B. bufo* development (*Baier et al., 2016*). However, the current findings are in contrast to findings of elongated tails of *Rana sylvaticus* treated with 2.9 mg a.e. L$^{-1}$ VisionMax (*Navarro-Martín et al., 2014*). In any case it appeared that these glyphosate-based herbicides interfere with molecular signaling processes or growth factors influencing tail growth. Clearly, detailed studies are needed to investigate the underlying mechanisms. Another explanation for a reduced tadpole growth after herbicide addition could be an herbicide-induced decrease of algae diversity that are an important food source for tadpoles (*Wojtaszek et al., 2004*). Also, in our current experiment algae were the preferred food for tadpoles although they were offered ad libitum ground fish food. Our finding that herbicide concentrations affected tadpole growth only at 15 but not at 20 °C suggests that increasing temperature did not significantly enhance the toxicity of the tested herbicide. This observation may also indicate heat hardening in the 20 °C groups, perhaps caused by an induction of chaperones and following stabilization of developmental processes (*Díaz-Villanueva, Díaz-Molina & García-González, 2015*). Overall, our findings indicate that temperature stress and herbicide stress do not necessarily point in the same direction (*Baier et al., 2016*; *Rohr, Sesterhenn & Stieha, 2011*). Not much is known on ecological consequences of reduced tl for tadpoles. However, it has been suggested that shorter tailed tadpoles can

escape less easily from predators than longer tailed ones (*Hoff & Wassersug, 2000*; *Wilbur & Semlitsch, 1990*). Also, tadpole tails are important energy and calcium reservoirs for metamorphosis eventually resulting in smaller toads (*Vitt & Caldwell, 2013*).

As expected, and in accordance with other studies (*Derakhshan & Nokhbatolfoghahai, 2015*; *Sanuy, Oromí & Galofré, 2008*), we found that tadpoles at 20 °C developed and grew much faster than at 15 °C, with tl being 1.6, bl 1.4 and bw 1.4 times higher on average. Higher temperature also accelerated tadpole development with tadpoles at 20 °C reaching Gs 35, whereas individuals at 15 °C reached on average Gs 29 until the experiment was terminated; bc (i.e. bl to bw ratio) was slightly increased by 5% at higher temperatures. Temperature also had a significant effect on the occurrence of deformed tails, as they were only observed on herbicide treated specimens at 15 °C but absent at 20 °C. Although climate change models predict higher temperatures during breeding periods (*Laloë et al., 2014*), breeding and hibernation phenologies of ectotherms might shift to earlier dates during warmer springs for compensating the warmer climate (*Benard, 2015*; *Gao et al., 2015*; *Klaus & Lougheed, 2013*). In some cases, the phenological shifts following warmer springs and winters even resulted in lower temperatures for the developing tadpoles (*Benard, 2015*). Overall there is still a great lack of data regarding seasonal variations in the water temperature of aquatic bodies used as spawning bonds and to what extent global climate change might affect these.

The ecologically most significant results of the current study are perhaps the interactive effects between herbicide and temperature on various parameters regarding tadpoles and algae. To our knowledge it is the first time that such herbicide-temperature interactions for a glyphosate-based herbicide was studied on European toad species and their associated algae. In the herbicide treated tubs at 15 °C 34% of all tadpoles had deformed tails, whereas no deformed tails occurred in the herbicide-free control at 15 °C or in all herbicide concentrations at 20 °C. Generally, deformation rates > 5% are considered unnatural and are therefore implemented amphibian monitoring guidelines (*Wagner et al., 2014*). However, in our experiment tail deformations were not related to herbicide concentrations; to what extent other principles than a dose-response relationship might be effective would need further studies perhaps by also using a greater sample size (*Vandenberg et al., 2012*). Substantial tail deformations by 50–90% due to glyphosate-based herbicides have also been found for the tropical frog species *Scinax nasicus* when exposed to 3.0–7.5 mg L$^{-1}$ of Glyfos (*Lajmanovich, Sandoval & Peltzer, 2003*). Also herbicides containing other active ingredients than glyphosate, e.g. isoproturon, have also been reported causing deformations in larvae of *Bombina variegata* and *Rana arvalis* at rather low concentrations (0.1–100 µg L$^{-1}$) (*Greulich, 2004*). Interestingly, we did not observe tail deformations on tadpole larvae in a previous experiment testing non-target effects of Roundup LB Plus on *B. bufo* (*Baier et al., 2016*). This putative discrepancy of effects might be due to the different active ingredients of the tested herbicide products, isopropylamin salt in Roundup LB Plus and potassium salt in Roundup PowerFlex which have been shown to evoke different non-target effects (*Cuhra, Bøhn & Cuhra, 2016*) or be the result of different, not-declared adjuvants in

the products (*Cox & Surgan, 2006*; *Cuhra, Traavik & Bøhn, 2013*; *Mullin et al., 2015*; *Relyea, 2005b*).

Interactive effects between herbicide and temperature resulted in a herbicide-induced decrease of bl, bw and tl by 2–10% at 15 °C, but the same parameters seemed to be (not significantly) increased by 2–8% at 20 °C for tadpoles in herbicide treated tubs compared to non-treated tubs. Overall, very little is known about combined effects of pesticides and temperature on aquatic organisms, especially amphibians (*Holmstrup et al., 2010*; *Rohr, Sesterhenn & Stieha, 2011*; *Wagner & Lötters, 2013*), as most studies investigated herbicide effects at one temperature level only. A study using Roundup Original MAX found effects on tail depth and body depth but not on the bl of *Rana pipiens* and *R. sylvaticus* tadpoles at 12 °C (*Relyea, 2012*). Similar to our findings snout-vent length of *Rana pipiens* larvae decreased when treated with 0.6 and 1.8 mg a.e. L$^{-1}$ Roundup Original and Roundup Transorb at 20 °C (*Howe et al., 2004*). In contrast with our study, growth rates of *R. clamitans* tadpoles treated with 1.43 mg a.e. L$^{-1}$ Vision herbicide at 20 °C increased (*Li, Cohen & Rohr, 2013*). Again, an explanation for these contrasting findings could be that these glyhosate-based herbicides usually differ considerably in the formulation of their active ingredients (*Cuhra, Bøhn & Cuhra, 2016*) their not-declared adjuvants that have been shown to be as detrimental to non-target species as the active ingredient (*Mann & Bidwell, 2001*; *Relyea, 2005a*; *Wagner et al., 2013*). To what extent spawning ponds used by amphibians are contaminated with glyphosate-based herbicides is very much dependent on regional circumstances, e.g. how careful herbicide application was conducted following regulations and distances to water bodies. Generally, most studies reporting glyphosate concentrations of surface waters considered larger lakes or streams, while many amphibians are using small, shallow and non-flowing water bodies for spawning. Nevertheless, reported glyphosate concentrations found in surface waters shortly after application scored between 0.27–3.10 mg a.e. L$^{-1}$ (*Wagner et al., 2013*).

Algae diversity (Shannon- and evenness-index) in our experiment was significantly reduced by herbicides and increased temperatures, whereas herbicide effects were more pronounced at higher temperature than at lower temperature (herbicide × temperature interaction). Reductions of algal cells after herbicides application were lower as expected by a pesticide supposed to kill plants. Possible effects of glyphosate-based herbicide formulations on aquatic algae communities have rarely been investigated and our study appears to be among the few addressing this topic for algae communities. Growth of green algae (*Scenedesmus acutus, S. quadricauda*) was inhibited by technical grade glyphosate and Ron-do, a glyphosate formulation used in aquatic environments (*Sáenz et al., 1997*). In toxicity tests analyzing the effect of glyphosate based herbicides on single species of diatoms and green algae, stronger negative effects on the diatom species (*Skeletonema costatum*) were observed (*Tsui & Chu, 2003*). Other studies found growth of *S. quadricauda* was inhibited when treated with an unspecified glyphosate-based formulation (*Wong, 2000*) or by treating four freshwater phytoplankton species with technical grade glyphosate (*Vendrell et al., 2009*). Since algae are able to produce mycosporine-like amino acids when stressed and this process can depend on the shikimate

pathway (*Shick & Dunlap, 2002*), it is possible, that glyphosate-based herbicides lead to a decrease of the build up of these amino acids (*Shick et al., 1999*), probably also increasing the effects of temperature stress on algal communities. As expected in our study, dissolved oxygen level was negatively correlated to water temperature. This means that the colder the water, the higher the oxygen content becomes (*Wetzel, 2001*). Although values of water oxygen concentrations varied broadly across temperature and herbicide treatments, we ruled out a lack of oxygen, as the lowest value measured being 8.2 mg $L^{-1}$, which is considered to be sufficient for proper development of tadpoles (*Ultsch, Bradford & Freda, 1999*). As a consequence of increased temperature dissolved oxygen levels decreased, which also affected tadpole growth. Contrary to our study, others found decreasing water pH levels with increasing Roundup concentrations (*Tsui & Chu, 2003*); however, this could also be due to unknown adjuvants in the different formulations used.

## CONCLUSIONS

Our findings might contribute several additional aspects to the broad discussion on non-target effects of pesticides. First, as extreme temperature events will most likely become more frequent due to human-induced climate change (*IPCC, 2013*), it seems imperative to address non-target effects of pesticides also in a climate-change context (*Köhler & Triebskorn, 2013*; *Rohr & Palmer, 2013*). The herbicide-temperature interactions in the current and a previous (*Baier et al., 2016*) experiment suggest that interspecific and intraspecific relationships in aquatic ecosystems will most likely be affected when taxa respond differently to these environmental stressors. Second, chronically sub-lethal effects in the morphology and body deformations as well as changes in algae communities such as observed in the current study and their ecological consequences should attract more attention. Third, our study also suggests that ecotoxicological protocols assessing non-target effects of pesticides conducted under constant temperature levels might underestimate real world conditions, especially for poikilothermic vertebrates.

## ACKNOWLEDGEMENTS

We are grateful to Mathias Jedinger for sharing his experience during the experimental phase and to Andreas Kitzler (Agilent Technologies Austria) for giving us extensive technical input and support for glyphosate analyses. The manuscript was improved by incorporating comments and suggestions of three anonymous reviewers.

### Funding

This study was funded by the Austrian Ministry of Agriculture, Forestry, Environment and Water Management (BMLFUW, project no. 100977). The funders had no role in study design, data collection and analysis, decision to publish, or preparation of the manuscript.

## Grant Disclosures
The following grant information was disclosed by the authors:
Austrian Ministry of Agriculture, Forestry, Environment and Water Management (BMLFUW): 100977.

## Competing Interests
The authors declare that they have no competing interests.

## Author Contributions
- Fabian Baier conceived and designed the experiments, performed the experiments, analyzed the data, wrote the paper, prepared figures and/or tables, reviewed drafts of the paper.
- Edith Gruber conceived and designed the experiments, performed the experiments, wrote the paper, prepared figures and/or tables, reviewed drafts of the paper.
- Thomas Hein analyzed the data, wrote the paper, reviewed drafts of the paper.
- Elisabeth Bondar-Kunze analyzed the data, wrote the paper, reviewed drafts of the paper.
- Marina Ivanković analyzed the data, reviewed drafts of the paper.
- Axel Mentler contributed reagents/materials/analysis tools, wrote the paper.
- Carsten A. Brühl analyzed the data, wrote the paper, reviewed drafts of the paper.
- Bernhard Spangl analyzed the data, reviewed drafts of the paper.
- Johann G. Zaller conceived and designed the experiments, performed the experiments, wrote the paper, prepared figures and/or tables, reviewed drafts of the paper.

## Animal Ethics
The following information was supplied relating to ethical approvals (i.e., approving body and any reference numbers):

This study was carried out in strict accordance with the recommendations in the Austrian animal experimentation law (Tierversuchsgesetz 2012, BGBl. I Nr. 114/2012) with a permission from the Austrian Federal Ministry of Science, Research and Economy (permit number BMWFW-66.016/0002-WF/V/3b/2015).

## Field Study Permissions
The following information was supplied relating to field study approvals (i.e., approving body and any reference numbers):

Toad eggs were sampled with permission of the pond owner (Vienna Municipal Department 49—Forestry, permit number MA49-808754/2014/3) and the respective authority for nature conservation (Vienna Municipal Department 22—Environmental Protection, permit number MA22-1629490/2014).

## Data Deposition
The raw data has been supplied as Supplemental Dataset Files.

## Supplemental Information

Supplemental information for this article can be found online at http://dx.doi.org/10.7717/peerj.2641#supplemental-information.

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
