# Peer review of "Non-target effects of a glyphosate-based herbicide on Common toad larvae (Bufo bufo, Amphibia) and associated algae are altered by temperature"

_PeerJ, doi:10.7717/peerj.2641_

## Round 0.1 · original submission · Major Revisions

Please revise, taking into consideration all comments of the reviewers.Although you must address all of their concerns, it is particularly important to pay attention to their statistical concerns. Please be advised that your manuscript may be sent out for review again after revision.

·

Basic reporting

No comments.

Experimental design

The temperature difference between 15 degrees and 20 degrees Celsius may have been to close to document a difference in chemical and morphological changes. Explain why these temperatures were selected. Also toads are not as sensitive to chemicals as other frog (Rana) species. Explain why toads were selected (besides the fact that they are commonly found in the area).

Validity of the findings

No comment.

Additional comments

This was a very interesting study involving several variables and climate change. More studies like this should be conducted.

Reviewer 2 ·

Basic reporting

The ms is very clearly written, the literature is well referenced and the figures are of high quality. Nevertheless, I have some comments:
- Repeatedly, the authors mention that they did not observe a clear relationship between herbicide concentrations and tadpole growth or algae density or diversity (lines 39ff, 276f, 332f. This statement, however, only refers to those tubs which contained glyphosate and not to those who contained zero glyphosate (controls). Such statement thus may be misleading. The authors should be more precise in saying that the presence of glyphosate in fact is decisive for an effect while increasing the concentration of glyphosate did not change this effect. Thus, the lowest tested glyphosate concentration, in these cases, is also the LOEC.
- The Introduction should point to the fact that interactions of pesticide and global warming effects have been predicted and outlined as a major point of concern for wildlife (Köhler & Triebskorn (2013): Science 341, 759-765; Rohr & Palmer (2013): Conserv Biol 27, 741-751 [the latter already cited, but in another context]. Regarding interactions of glyphosate and temperature, the authors also have missed the recent paper of Thai Hoan Nguyen et al. (2016): Pest Management Sci 72, 1031-1039.
- Intro, line 81: better use “… other sublethal effects” because growth and a number of developmental parameters (mentioned in line 79) are also sublethal. In line 104: spelling mistake “herbicide”.
- Material and Methods, line 176: spelling mistake “Petri dish”. Line 181: “similar amounts” is not precise. Can this be further specified?
- Results, lines 244ff: The text does not correspond to Figure 2 here. Tail length is significantly affected only at 15°C (not generally), tail deformations at 15°C lack statistics in Fig 2 (though mentioned to show significance in the text), and the body length was significantly affected by glyphosate at 15°C, as shown in Fig 2 (contrary to the statement in the text). In general, the letters indicating significance are missing in Figures 2B and 2F (at least the data presented fin Fig 2B may the reader speculate about significance).
- Results, lines 277ff: There is a logical break here. One sentence refers to the absence of malformations in the control group individuals. The next sentence starts with “Most of them” but refers to malformed tadpoles. Better start with “Whenever tail deformations occurred…” or similar. In line 288, please add a comma each after “was” and after “treatments”. In line 299, please change “at treatments” into “in the treatments”.
- Results, para lines 314-322: To my opinion, the Results section should start with this para as it gives basic information on the reliability of the experiments but, I admit, this is a question of personal spelling style.
- Discussion, line 340: I do not think that the data are actually “in contrast” to those referenced here since both studies revealed developmental effects on tail growth indicating interference with molecular signaling processes or growth factors. Lines 345ff: The sentence is somewhat awkward, perhaps the authors mean that temperature stress and pesticide stress do not have necessarily to point into the same direction.
- Discussion, lines 359f: This observation speaks in favor of heat hardening in the 20°C groups, likely caused by an induction of chaperones and following stabilization of developmental processes. Such mechanistic interpretation should be mentioned here, at least shortly. Information on such processes can be obtained from Fernando Diaz-Villanueva et al. (2015): Int J Mol Sci or numerous other review papers on stress/heat shock proteins published previously.
Discussion, lines 392ff (“Overall, very little is known…”): Knowledge on combined temperature and pesticide effects has been summarized in the review paper of Holmstrup et al. (2010): Sci Tot Environ 408, 3746–3762.
Discussion, lines 441f: This demand has been raised by Köhler & Triebskorn (2013): Science 341, 759-765 before, thus may be referenced here.

Experimental design

No comments

Validity of the findings

No comments

Additional comments

Great study, I think. The ms just needs minor revision.

Reviewer 3 ·

Basic reporting

The Manuscript entitled "Non-target effects of a glyphosate-based herbicide on Common Toad larvae (Bufo bufo, Amphibia) and associated algae are altered by temperature" details a laboratory (mesocosm) experiment investigating the effects of different concentrations of the herbicide Roundup PowerFlex and two different temperature regimes on the development of tadpoles of the common toad, and associated algae communities. The authors tested whether herbicide concentrations affected the development of larvae and algae communities and how temperature alters potential effects of the herbicide formulation. The authors found out that herbicide contamination significantly affected different biological parameters of tadpoles (larval development, body length, body width, tadpole mortality), especially at 15°C. Individuals exposed to the same concentrations at 20°C showed much better biological parameters, probably due to a shorter exposure period conditioned by a faster development due to higher temperatures. Similarly, while no malformations were observed for tadpoles exposed to Roundup PowerFlex at 20°C, 76% of the tadpoles exposed to this pesticide at 15°C exhibited deformed tails. This leads to the conclusion that higher temperature regimes may ameliorate the effects of pesticide exposure for amphibian larvae.
Furthermore, the Authors detected a reduction of the associated algae diversity induced by herbicide contamination, at both temperature regimes.
The article is written in good and understandable English and includes sufficient introduction and background for the reader know how it fits into the current field of knowledge. All figures provided are relevant to the manuscript and have appropriately been described. All appropriate raw data is available.
Concerning the effects of pesticide applications on non-target organisms, I feel this would be an important work to publish if some of the major concerns mentioned below are addressed.

Experimental design

The primary research of the article lies within the scope of the journal. The research questions are well and clearly defined, and are relevant within the field of non-target organism exposure to pesticides. The experimental design is sound and the methods are sufficiently described in order to be reproducible by other scientists. All ethical standards have been met.

Validity of the findings

Regarding statistical evaluation, I strongly recommend including all test statistics for all conducted analyses in the manuscript. As it is now, the reader has now way to determine which results were significant, and at which significance level. Furthermore, statistics should include comparisons of the measured biological parameters within groups (temperature regimes), instead of only between groups. Considering that the results regarding the different Roundup concentrations were rather ambiguous, this may help to better understand how the different concentrations related to each other within a specified temperature.
Furthermore, I feel like a sample size of 10 Individuals per replicate, may have helped in making the analyses more robust, and should not have caused a much greater workload. However, I don't regard this as a reason to not accept the manuscript.

Secondly, the section concerning malformation rates of larvae has to be revised. Lacerated tails are not considered as actual developmental malformations, neither by Cooke (1981) nor Bantle et al. (1998). This kind of injuries most probably originates from intraspecific competition / "predation". As a consequence, they should be removed from that analysis.

See major Comments (below)

Additional comments

Major comments
I feel it is imperative that the authors include all test statistics conducted for the mentioned analyses. As it stands now, it is impossible for the reader to determine which results were significant, and at which significance level (the only information available is that alpha was set to 0.05).

Related to this aspect, I think it would be very important to add the statistical data for the tests conducted within temperature groups (e.g. test significances for the tested parameters in the different pesticide concentrations, within a temperature regime). Statistics are only provided for comparisons between groups (temperature) in the manuscript itself. As there is no clear trend in the biological responses between tested concentrations within a temperature regime, this data would be very useful in order to interpret the results more clearly.

Furthermore, I feel that the section concerning malformation rates has to be revised before the manuscript can be accepted. The authors defined two types of observable malformations in tadpoles: “lacerated tails” and “curved tail tips”. While I agree with the classification of the latter, the former cannot be considered an actual malformation. Neither of the two mentioned works in this study regarding the identification of malformations in tadpoles (Cooke 1981 and Bantle et al. 1998) recognized this deformation as actual developmental malformation. This kind of injury is often caused by intraspecific competition / ”predation” and has been observed under laboratory conditions. Thus, individuals classified in this group should be eliminated from the malformation analyses.

Minor comments
Introduction
Line 81: Please name some examples of the sub-lethal effects found, and how they may affect the studied organisms.
Line 117 – 120: This strongly depends on how you define a “stressor”. An increased temperature by itself will probably not create additional stress for the tadpoles (at least not on a level where it would be measurable to a significant extent), but enhance the development of the tadpoles (as you mentioned before). For temperature to be considered as “stress”, it would have to be linked some kind of negative effect on the organism, to which it has to respond to. In the wild, this would usually be caused by desiccation of breeding ponds. It is known that amphibians respond with increased development in order to survive such events.
Material and Methods
Line 138 – 142: You write you added 1.5mg a.e. L-1 at two instances, with an additional mg a.e. 5 and 11 days after starting the experiment. As I read it, you have three additional applications, which are added in 2 instances. Were the two last concentrations added together? If so, why are they reported as 1.5mg and 1mg a.e. instead of 2.5mg a.e.?
Line 157 – 164: Why didn’t you go for 10 eggs per replicate? Although you have 25 individuals per concentration considering the 5 replicates, increasing the number of eggs would make the analyses statistically more robust.
Line 177 – 179: Lacerated tails as documented in Figure 1b are not considered as malformations, neither by Cooke (1981) nor Bantle et al. (1998). These kind of damages mostly result from Tadpoles occasionally feeding on their conspecifics (commonly nagging at the tails), which also occurs when they have enough food at their disposal. It is important to differentiate between actual malformations and damages caused by external factors. I would suggest revisiting all cases of “lacerated tails” and only include those cases in which external factors can be excluded. Even in Cooke (1981), the only similar malformation reported is “damage” to the tip of the tail, which does not correspond to the here reported observation. I agree that Fig 1c is a clear case of malformation, though.
Line 192: add a comma after present.
Line 232 – 237: Please always define which co-variables were used for which ANCOVA model.
Results
Line 244 -322: For all results, please report the statistical information for the parametric tests (e.g., F statistics, degrees of freedom, and p-value for the ANCOVAs and correlations).
Line 246: I believe you never mentioned how the body-condition was calculated. Please report the calculation method, as there are plenty of different ways to calculate it, and include it in the material & methods department (going by Fig. 2 you used a length to width ratio measurement).
Line 256 – 260: While I agree that this Temp x Herbicide combination resulted in these reduced tail lengths at 15°C, you make it sound as if at 20°C, it caused an increase in TL when compared to reference individuals, while it would seem more logical that there is simply no effect at all. Were there any significant differences between the tested groups (0, 1.5, 3, 4, 4p) at 20°C? You also mention that TL was about 8% shorter for exposed individuals when compared to control groups. Was the test statistic significant? Biological parameters have great variation ranges, even within individuals of the same populations. In any case, I would highly suggest testing for significant differences within the Temperature groups if it was not done, and including the results in the text in order to make it less ambiguous.
Line 262 - 273: As mentioned above, I believe it would be important to report whether the differences between tested concentrations were significantly different within the two temperature groups.
Line 274 – 279: As it stands, the reported reference image is not a malformation, but most likely caused by an external influence (probably by “predation” by conspecifics).
Line 297 – 300: First you write that no significant effect of Temperature x Herbicide could be observed on pH and oxygen. Then you write that the pH in the control at 15°C was significantly lower than with herbicides, with similar results for 20°C (lowest value in the control, highest for the pulse treatment). Then, the same trend is reported for oxygen concentration. This is a contradiction. Please specify/correct the statement.
Discussion
Line 337 – 340: As mentioned before, it would be highly interesting to know whether these measured effects are significant or not. While a reduction of TL can be indeed observed when looking at Fig. 2a, it is also true that the results fluctuate considerably between concentration groups. As biological values often tend to show such patterns, a sample size of 10 individuals per replicate could have helped to reduce this fluctuation. I think it is rather important to include the actual statistics for the replicates within the two temperature classes.
Line 350 – 352: The tails are also used as energy and calcium reservoir for metamorphosis. Thus, shorter tails can have an impact on metamorphosis itself, as the individuals may have less energy reserves, resulting in smaller toads.
Line 360: Is there a possible explanation as to why malformations only occurred at 15°C? 5% malformation rates are seen as natural in pristine environments (although the actual rate is probably higher). It is striking that no malformations were observed for any of the tadpoles in the 20°C group.
Line 372 – 378: As mentioned before, please revise all malformations declared as “lacerated tails”, as there is no such classification, neither in Cooke (1981) nor Bantle et al. (1998), and the most probably stem from intraspecific “predation”.
Line 376: It is possible that the variation in biological responses is too great to measure with a sample size of “only” 25 individuals per replicate (10 individuals per replicate may have helped reduce the variance), and that a greater sample size is needed in order to detect dose-response correlations.
Line 388: I suggest including Cox & Surgan 2006. Unidentified inert ingredients in pesticides: Implication for Human and environmental health. Environmental Health Perspectives 114. 1803 – 1806.
Line 394 – 396: This sentence feels out of place, and should probably be moved to line 392 (before you mention the lack of data regarding temperature and pesticide synergies). Also, the citation is lacking an author  “(2012)”
Line 410: “was” is wrong here  were. Nevertheless, I`d suggest using a more appropriate word like scored.
Line 414: Not that surprising if you consider that the herbicide was formulated to kill terrestrial plants (weeds). While algae are still plants, this roundup formulation was not conceived for aquatic ecosystems (the environment is totally different), and as you yourself mentioned, very little data concerning this topic is available.
Line 420: place a comma after “species of diatoms and green algae”.
Lines 422: “was” inhibited
Lines 428 – 430: This sentence is difficult to read. Maybe like this: “As expected in our study, dissolved oxygen level was negatively correlated to water temperature. This means that the colder the water, the higher the oxygen content becomes”
Line 433: While the amount of oxygen is enough for tadpoles to develop, is it perhaps possible that increased availability also increases the development process of the individuals? If this is the case, it could explain why you had such high variations for all of your parameters, even within the testing groups.

---

## Round 0.2 · Minor Revisions

Thank you for your careful attention to the reviewers' comments. I have just a few very minor changes I would like you to make before your paper is accepted, as follows:

Lines 274-5: concentrations neither affected
Line 279: however it
Line 417: detailed studies are needed
Line 546: manuscript was improved
Finally, please make sure that any new citations you added in the text are also in the References Cited list.

---

## Round 0.3 · accepted · Accept

Thank you for making the changes I requested. I'm happy that we can now accept your paper for publication.